# Detecting Mild Water Stress in Olive with Multiple Plant-Based Continuous Sensors

**DOI:** 10.3390/plants10010131

**Published:** 2021-01-11

**Authors:** Giulia Marino, Alessio Scalisi, Paula Guzmán-Delgado, Tiziano Caruso, Francesco Paolo Marra, Riccardo Lo Bianco

**Affiliations:** 1Department of Plant Sciences, University of California, Davis, CA 95616, USA; pguzmandelgado@ucdavis.edu; 2Department of Agricultural, Food and Forest Sciences (SAAF), University of Palermo, 90133 Palermo, Italy; alessio.scalisi@agriculture.vic.gov.au (A.S.); tiziano.caruso@unipa.it (T.C.); francescopaolo.marra@unipa.it (F.P.M.); 3Agriculture Victoria, Department of Jobs, Precincts and Regions, Tatura, VIC 3616, Australia

**Keywords:** fruit gauge, *Olea europaea* L., precise water management, stem water potential, turgor pressure, water relations

## Abstract

A comprehensive characterization of water stress is needed for the development of automated irrigation protocols aiming to increase olive orchard environmental and economical sustainability. The main aim of this study is to determine whether a combination of continuous leaf turgor, fruit growth, and sap flow responses improves the detection of mild water stress in two olive cultivars characterized by different responses to water stress. The sensitivity of the tested indicators to mild stress depended on the main mechanisms that each cultivar uses to cope with water deficit. One cultivar showed pronounced day to day changes in leaf turgor and fruit relative growth rate in response to water withholding. The other cultivar reduced daily sap flows and showed a pronounced tendency to reach very low values of leaf turgor. Based on these responses, the sensitivity of the selected indicators is discussed in relation to drought response mechanisms, such as stomatal closure, osmotic adjustment, and tissue elasticity. The analysis of the daily dynamics of the monitored parameters highlights the limitation of using non-continuous measurements in drought stress studies, suggesting that the time of the day when data is collected has a great influence on the results and consequent interpretations, particularly when different genotypes are compared. Overall, the results highlight the need to tailor plant-based water management protocols on genotype-specific physiological responses to water deficit and encourage the use of combinations of plant-based continuously monitoring sensors to establish a solid base for irrigation management.

## 1. Introduction

Climate change and water scarcity, along with an intensification of agricultural systems, are urging the reduction of irrigation volumes applied in agricultural systems. Intensive olive orchards (300–600 tree ha^−1^) are more productive per surface unit with respect to traditional systems (less than 300 tree ha^−1^), but they are more sensitive to water deficit. Hence, plant-based irrigation management is a key strategy to maintain economical profitability and environmental sustainability in intensive systems [1].

Among the different indicators of plant water status, stem water potential (Ψ_stem_) is currently the most widely studied and used. Thus, crop-specific protocols are available in the literature to manage irrigation in fruit tree systems using midday Ψ_stem_—the Ψ_stem_ at solar noon—as an indicator of water stress [2,3]. In olive, two recent studies developed in two different environments (Sicily and Argentina) and with multiple cultivars (Arbequina, Nocellara del Belice, and Olive di Mandanici) suggested −2.0 MPa as a midday Ψ_stem_ threshold for mild water stress since a clear and gradual decrease in stomatal conductance and net photosynthesis occurred at Ψ_stem_ below this value [4,5]. However, when Ψ_stem_ was above −2 MPa, the high variability of gas exchange was not reflected in Ψ_stem_ changes. This is probably associated with the variable near-isohydric behavior of some olive cultivars that influence the degree of Ψ_stem_ changes at low water stress levels [4,6]. Thus, alternative indicators to midday Ψ_stem_ are needed to improve the precision of water stress detection at Ψ_stem_ ≥ −2 MPa and maximize water productivity in olive orchards.

In recent years, thanks to the rapid advancement of technologies, various plant-based sensors have been tested and proposed for automated water management in orchard systems. Most of these sensors do not measure plant water status directly but they monitor specific physiological processes that correlate in different ways and degrees with plant water status. Since tree physiological responses are mediated by several factors, including tree phenological stage, environmental conditions, or genotype-specific traits, the development of simplified and uniform water management protocols using these sensors is complex [7,8]. As an example, the use of the leaf patch clamp pressure (LPCP) probe [9] could be very promising for irrigation scheduling in ‘Arbequina’ olive orchards [10,11]. This sensor measures the variability in the output pressure (*P*_p_, kPa) between two magnets clamped to a leaf. When trees are not water-stressed, *P*_p_ is a good proxy of the attenuation of turgor pressure (*P*_c_), with *P*_p_ and *P*_c_ being inversely related. For instance, when leaf turgor pressure decreases in response to daytime stomatal opening, *P*_p_ gradually increases while when stomata close at night and turgor pressure increases, *P*_p_ gradually decreases (phase 1). However, when trees enter water deficit, a clear decrease of *P*_p_ values associated with the accumulation of excess air in leaf tissues occurs, resulting in a semi-inversion of the *P*_p_ diurnal trend, with a depression in the middle of the day (around solar noon, phase 2) [12]. More severe stress will cause a complete inversion of the *P*_p_ curve, with high peak in the night hours and lower values in the day hours (phase 3). An example of the above mentioned phases of inversion is reported in Figure 1.

This change in the shape of the diurnal *P*_p_ curve was associated with a midday Ψ_stem_ < −1.2 MPa for the cultivar Arbequina grown in Southern Spain [10,11]. However, when this sensor was tested on different olive cultivars and environments, an evident change in the shape of the *P*_p_ curve was only clearly observed at midday Ψ_stem_ < −2.5 MPa [13]. ‘Arbequina’ is an olive cultivar selected in temperate regions and has shown a lower capability to adapt to drought, and a higher bulk elastic modulus (ε) (i.e., a higher tissue rigidity) than other cultivars [14]. A high ε is generally associated with a low capability of the cells to maintain turgor, which probably influenced the high sensitivity of the *P*_p_ daily curve to water stress in this cultivar.

Some recent studies suggested combining the use of different plant-based sensors to improve water stress detection in olive. Scalisi et al. [7] reported that combining leaf and fruit sensing improves water stress detection in two olive cultivar. Girón et al. [15] investigated physiological responses of olive fruits under different levels of water deficit (with midday Ψ_stem_ < −2.5 MPa) and found that, at moderate water stress, fruits are weaker water sinks than leaves, as opposed to what is observed in other species [16,17]. Yet, when the severity and duration of water stress increases, fruit sink activity may become stronger than leaf sink activity for water. Rodríguez-Domínguez et al. [18] combined stem sap flow and leaf turgor pressure measurements and demonstrated that different indicators should be used depending on the intensity of stress, with sap flow being a potential solution to detect severe water stress when the *P*_p_ curves are inverted. All these studies highlight the complexity of the detection of water stress in olive, particularly for light to mild water stress conditions and when different cultivars are considered. Therefore, a more comprehensive characterization of water stress onset is needed for the development of automated irrigation protocols aiming to increase olive orchard environmental and economic sustainability.

The main aim of this study was to determine whether a combination of continuous leaf turgor, fruit growth and sap flow responses improves the detection of mild water stress in olive—i.e., when trees approach a midday Ψ_stem_ threshold of −2.5 MPa, before the inversion of the *P*_p_ curves. In addition, two Sicilian olive cultivars (Olivo di Mandanici and Nocellara del Belice) were investigated to assess if the combined use of these physiological indicators could provide useful information, regardless of the cultivar-specific response to water deficit. In particular, ‘Olivo di Mandanici’ (OM) showed a slower decrease in Ψ_stem_ and a lower tendency to show semi-inversion or inversion of the *P*_p_ curves in response to water withholding with respect to ‘Nocellara del Belice’ (NB) [7,13,19], resulting in lower irrigation needs.

The first hypothesis was that a combination of physiological parameters provides better indications of mild water stress than individual indices. The second hypothesis was that physiological responses to stress may differ in cultivars with different sensitivity to drought, thus requiring the use of alternative indicators to assess mild water stress.

## 2. Materials and Methods

### 2.1. Orchard Characteristics

The experiment was carried out in an intensive olive orchard located near Sciacca (37′32″ N, and 13′02″ E, 150 m a.s.l.) in Southwestern Sicily (Italy) during August and September 2015.

The soil is a sandy clay loam (60% sand, 18% silt, 22% clay), with pH 7.7 and active carbonates lower than 5%. The orchard was planted in 2012 with seven different Sicilian genotypes previously selected by the Department of Agricultural, Food and Forest Sciences at the University of Palermo. The density was 666 trees ha^−1^, with a spacing of 5 m between rows and 3 m on North-South oriented rows. Trees were trained to free palmette with a single trunk and three scaffolds: the first at 0.5–0.6 m from the ground, the second at 1.30–1.40 m, and the third at 2.10–2.20 m. All trees received the same conventional cultural cares from planting until the beginning of the current experiment.

### 2.2. Environmental Conditions

The climate is Mediterranean, with an average yearly precipitation of 516.2 mm and an average monthly temperature of 18.2 °C (period 1965–1994, meteorological station of Sciacca, Servizio Informativo Agrometerologico Siciliano). Data of average daily temperature and average daily relative humidity used to calculate vapor pressure deficit (VPD), as well as total daily precipitation were obtained from the meteorological station of Sciacca.

At the beginning of August, the weather was particularly hot and dry, with VPD values as high as 4 kPa (Figure 2). A rain event on 8 August, decreased the VPD below 2 kPa for about a week. VPD showed a constant pattern in the second half of August, with daily minimum values of ~0.5 and maximum values of ~2.5 kPa. At the beginning of September, a heavy rain event (>50 mm) caused a marked decrease in the daily maximum VPD values to ~1 kPa. A steep increase of VPD was then observed for the 5 days between 16 and 20 September.

### 2.3. Irrigation Management and Experimental Design

Irrigation water was supplied by two self-compensating in-line drippers per plant delivering 16 L h^−1^. In mid-July, when pit hardening usually occurs in the experimental area, trees were not irrigated until their midday Ψ_stem_ decreased below −2.5 MPa. The main aim was to keep the trees under a mild level of stress (midday Ψ_stem_ between −2 and −2.5 MPa) based on the results of Marino et al. [4]. In September, when the oil accumulation rate is generally high, irrigation was applied to avoid water stress and keep the Ψ_stem_ above −2.0 MPa. The experimental design was composed of three blocks per cultivar made of three adjacent trees each and randomized within the row. The two cultivars were displayed in two separate rows. Measurements were performed in the central tree of each block. One sensor of each type was installed in each selected tree, for a total of 18 sensors (three sensor-types per three trees per two cultivars). The sampling procedure and number of measurements corresponded to the standards for these settings and type of investigation [7,19] and fully support the strength of the presented outputs.

### 2.4. Measurements

Plant water status was monitored weekly on nine trees per cultivar measuring midday Ψ_stem_. One current-year west-oriented shoot with five or six pairs of fully expanded leaves per selected tree was covered with plastic wraps and aluminum reflective foils at least 1 h before measurements to reduce leaf transpiration and equilibrate branch water potential with tree water potential [20] Shoots were then detached and Ψ_stem_ was measured using a pressure chamber (PMS Instrument Co., Model 600D, Corvallis, OR, USA).

Starting from 1 August, LPCP probes (YARA ZIM Plant Technology GmbH, Hennigsdorf, Germany), sap flow sensors (Ecomatik, UP-GmbH, Dachau, Germany), and fruit gauges based on linear variable displacement transducer (LVDT) sensors were installed on the selected trees to monitor relative changes in leaf turgor pressure, sap flow density (*Q*, mL cm^–2^ min^–1^), and micrometric fruit diameter variations, respectively.

The LPCP probes were installed following reported methodology [9,21] on fully expanded leaves of west-oriented, current-year shoots, similar to the ones used for Ψ_stem_. The output signals of the LPCP probes (*P*_p_, kPa) were standardized using z-scores [i.e., z = (x − mean)/standard deviation] to allow comparisons among leaves with different initial turgor pressure when sensors were mounted. The inversion of the *P*_p_ daily curve was used to characterize the following three stress phases based on [12]: phase 1 or no stress, the leaf is in a turgescent state characterized by an inverse relationship between *P*_p_ and *P*_c_, and midday Ψ_stem_ is above ~−2.0/2.5 MPa; phase 2 or mild stress, characterized by semi-inverted *P*_p_ curves and midday Ψ_stem_ below ~−2.0/2.5 MPa but above ~−3.0/−3.5 MPa; phase 3 or severe stress, the leaf is significantly dehydrated (unfavorable ratio of air to water in the leaf tissue) and midday Ψ_stem_ is lower than ~−3.0/−3.5 MPa (in this state, the *P*_p_ daily curve is inverted compared to phase 1, i.e., *P*_p_ reaches minimum values around solar noon and maximum values during the night). Reported midday Ψ_stem_ values for each phase were developed in a previous study performed in the same study orchard [13].

Granier-type heat dissipation probes [22] were used to measure sap fluxes. The sap flow probes were placed on the west side of the trunk at 50 cm from the ground and covered with aluminum foil to avoid the effect of direct sunlight on sensor temperature readings. The temperature signals from the probes were recorded at 30-min intervals using a CH6 data logger (GMR Strumenti Sas, Scandicci, Florence, Italy).

The sap flow density (*Q*, mL cm^−2^ min^−1^) was calculated from the temperature difference between the two needles (∆T) and the maximum value of ∆T (∆Tmax) at night as follows [22]:Q = 0.714 × [(∆Tmax − ∆T)/∆T]

Fruit gauges were installed on fruits located in the same main branches where the LPCP probes were installed, as close as possible to the leaf sensors, and connected to CR1000 data loggers (Campbell Scientific Inc., Logan, UT, USA) set to record micrometric variations (mV) at 15-min intervals. Data in mV were subsequently converted in μm starting from an initial fruit diameter measured with a digital caliper and considering that a change in 1 mV was equivalent to 3.3 μm. Fruit diameter data were subsequently smoothed by applying a filter using a 15-point convoluted spline to remove noise [23].

Relative growth rate (RGR, μm mm^−1^ min^−1^) was calculated using the following equation:RGR = (ln D2 − ln D1)/(t2 − t1)
where D1 and D2 are fruit diameter at times t1 and t2, respectively.

### 2.5. Statistical Data Analysis

Exploratory data analysis (EDA) was carried out to assess patterns and anomalies of physiological parameters in time series. Statistical analyses of the data were performed using R program [24]. Analysis of variance was performed with cultivar and time as main factors and Tukey’s multiple comparison test was used to separate means. Linear regression analyses between sensor output and Ψ_stem_ were performed and Student’s *t*-test was used to compare regression coefficients.

## 3. Results

### 3.1. Stem Water Potential

Midday Ψ_stem_ showed only slight differences between the two cultivars throughout the experimental period. In August, midday Ψ_stem_ values were lower than the threshold value of −2.0 MPa, mainly oscillating between −2 and −2.7 MPa (Figure 3). The Ψ_stem_ values were higher in Olivo di Mandanici (OM) than in Nocellara del Belice (NB), both at the beginning and the end of this month. In September, Ψ_stem_ increased to values above the −2.0 MPa threshold, with NB showing slightly higher values in the second half of the month.

### 3.2. Continuous Trends of Leaf Turgor, Sap Flow and Fruit Relative Growth Rate

A combination of continuous measurements of physiological parameters can provide insight into processes occurring when trees experience an increased water stress, as well as differences associated with cultivar-specific traits. Based on the EDA of the curve daily shape, no stress was detected through almost the entire experimental period, as highlighted by the green circles in Figure 4, indicating a normal, bell-shaped curve for the specific day (phase1 or no stress). Only on five days (27 August for OM, and 7, 19–20, and 27–28 August for NB) did the EDA of the curve daily shape identify some stress level, which was highlighted by yellow and red circles indicating partial or full inversions of the curves during these days (phases 2 and 3 of stress onset and high stress, respectively).

Despite such a low stress detection based on the *P*_p_ daily curves, a strong day to day variability in *P*_p_ dynamics can be clearly observed between two consecutive irrigation events (Figure 4). This is particularly evident during the second half of August, when daily minimum *P*_p_ values (*P*_p-min_) increased in both cultivars, with values ranging from about −1, just after the irrigation, to positive values recorded after five to seven days of water withholding (red continuous arrows in Figure 4). The cultivar OM also showed a progressive increase of maximum daily *P*_p_ values (*P*_p-max_) from ~0.5 up to ~2 between two consecutive irrigation events (red dashed arrows in Figure 4a). In contrast, *P*_p-max_ was relatively constant in NB, with approximate values of 0.5 to 1.0 (red dashed arrows in Figure 4b).

At the beginning of September, the increased frequency of irrigation re-established relatively constant *P*_p_ values for about two weeks, with *P*_p-max_ between 0 and 1 and *P*_p-min_ around −1. The cultivar NB had a more stable *P*_p-max_ and more variable *P*_p-min_, while OM had more stable *P*_p-min_ and more variable *P*_p-max_. Starting from mid-September, both cultivars showed a continuous increase of *P*_p-min_ and *P*_p-max_ through to the end of the season (Figure 4) most probably in response to the strong and rapid increase in VPD observed in this period (Figure 2).

Fruit RGR dynamics showed clear changes in the period between two irrigation events, with a progressive decrease of the daily minimum RGR value (RGR_min_), from values close to 0, just after the irrigation, to values of −0.3 μm mm^−1^ min^−1^ in OM and 0.15 μm mm^−1^ min^−1^ in NB (Figure 5, red continuous arrows). The cultivar OM also showed an increase of maximum daily RGR values (RGR_max_) between two irrigation events from ~0 up to 0.3–0.4 μm mm^−1^ min^−1^, while in NB, RGR_max_ was constant between two consecutive irrigation events (red dashed arrows in Figure 5). During all of September, RGR was close to 0 in both cultivars except for the period between 18 and 22 September, when fruits showed larger size fluctuations in response to 10 consecutive days without irrigation.

The dynamics of *Q* was less clearly affected than the previous parameters by the irrigation events (Figure 6). In August, OM daily *Q* dynamics showed a very high peak of transpiration rates during the morning (values reaching up to 0.025 L cm^−2^ h^−1^) but average values for the rest of the day ranged between 0.015 and 0.02 L cm^−2^ h^−1^. NB did not show a clear morning peak and average values in August were between 0.015 and 0.02 L cm^−2^ h^−1^. In September, during the 10 days of irrigation withholding, NB *Q* values increased from 0.018 to 0.024 L cm^−2^ h^−1^, while OM *Q* values decreased from 0.021 to 0.015 L cm^−2^ h^−1^.

### 3.3. Relationships between Stem Water Potential and Sensor-Derived Physiological Parameters

To gain further insight into the leaf, fruit, and sap dynamics, and evaluate their potential as mild water stress indicators, we analyzed the relationship between midday Ψ_stem_ and the minimum values of *P*_p_ (*P*_p-min_), the difference between the minimum value and the maximum value of RGR (RGR_range_), and the daily average of *Q* values (*Q*_ave_) (Figure 7). These three parameters or indicators were selected because they showed the highest variability between two consecutive irrigation events, as derived from the hourly data analyses (Figure 3, Figure 4 and Figure 5). *P*_p-min_ showed a significant linear and negative relation with Ψ_stem_ (Figure 7a), with no statistical difference observed between the slopes of the two tested cultivars (*p* = 0.3, Table 1). Similar to *P*_p-min_, the RGR_range_ was negatively correlated with Ψ_stem_ (Figure 7b) but the slope was significantly higher in OM than in NB (Table 1, *p* = 0.03). The different response of the two cultivars was also highlighted by the relationship between *Q*_ave_ and Ψ_stem_, which was significant, linear, and positive in NB and not significant in OM (Figure 7c), with significant difference between the slopes (*p* = 0.001, Table 1).

### 3.4. Hourly Dynamics of Leaf Turgor, Sap Flow, and Fruit Relative Grow Rate

To better capture the synchronism of the diurnal kinetics of water flow in stem, leaves, and fruits under mild stress as well as the differences between the two cultivars, we show a six-day snapshot, corresponding to a period between two consecutive irrigation events, from 15 to 21 August (Figure 8). Similar trends were observed in other periods between irrigation events along the course of the experiment. *P*_p_ was reported as 1 − *P*_p_ to make its changes directly related to changes in turgor pressure (*P*_c_) and fruit RGR. This allows an easier comparison of the dynamics of the different indicators.

The daily trends of *P*_p_, *Q*, and RGR were constant for the first two days of irrigation withholding and after that they started showing clear changes in their daily dynamics.

Based on the hourly dynamics of the three indicators under stress (from 17 to 21 August), four key timeframes were identified.

T1 (from ~6:00 to 10:00 h): stomata open and tree transpiration starts to increase until reaching a peak, which was more pronounced in OM than in NB. Leaf turgor decreased in both cultivars in response to stomatal opening while fruit RGR kinetics were more variable. OM showed an initial increasing trend of RGR in response to the beginning of transpiration followed by a sharp decrease to values of ~−0.2 μm mm^−1^ min^−1^, corresponding to the transpiration peak. NB showed milder and constant decrease of fruit RGR through the whole period.

T2 (from ~10:00 to 17:00 h), characterized by a midday depression of transpiration, accompanied by an apparent recovering of leaf turgor in NB and a slowdown of leaf turgor decrease rate in OM, and by a new increase of fruit RGR in OM.

T3 (from ~17:00 to 20:00 h), when transpiration decreased constantly until it reached zero. In this timeframe, in OM, a sharp increase in leaf turgor and fruit RGR was observed, while in NB, fruit RGR increased slightly at a slower rate reaching values around zero while leaf turgor decreased slightly.

T4 (from ~20:00 h to 8:00 h), during nighttime when stomata were closed, transpiration was absent. Fruit RGR of OM decreased reaching values close to zero and leaf turgor continued to increase but at a slower rate than in T3. In NB, fruit RGR was constant and leaf turgor values continued increasing similarly to OM.

## 4. Discussion

### 4.1. Use of Multiple Continuous Sensors Improves Mild Stress Detection

The analysis of the collected data confirmed our first hypothesis that a combination of physiological parameters provides better indications of mild water stress than individual indices. In fact, even if moderate-to-high water stress was not detected during most of the experimental period by the *P*_p_ trends, a commonly used indicator derived from the PLPC (i.e., no inversion was detected in the *P*_p_ daily curves), the relationships between continuously measured parameters clearly changed as soil water availability decreased between two irrigation events, highlighting physiological responses of both olive cultivars to reduced water availability.

For example, in the period between two irrigation events, *P*_p_ daily minimum values increased continuously, indicating a gradual decrease of the daily maximum leaf turgor pressure at night (Figure 4), since trees were not able to fully recover the water lost during the day [25]. The increase in *P*_p_ daily minimum values has been previously described as an indicator of mild water stress in olive [13], as also supported by the significant relationship we observed between this indicator and stem water potential (Figure 7a). Similarly, the RGR daily range increased strongly with irrigation withholding, and it was also highly correlated with tree water status (Figure 7b), highlighting a pronounced fruit shrinkage in response to water stress as previously reported in several crops [26,27,28,29,30] and, most recently, also in olive [7,31]. Changes in sap flow density fluxes between irrigation events were less noticeable than those in leaf *P*_p_ and fruit RGR, most probably because of the high impact that the environment (namely VPD) has on this parameter [32], further affected by tree water status [8,33]. However, both cultivars responded to drought stress by modifying the daily dynamic of transpiration rates, which showed a high peak in the morning after a few days of water withholding (Figure 6), as also previously revealed by stomatal conductance values [7]. This is a drought resistance mechanism since plants try to take advantage of favorable environmental condition in the morning (lower VPD and better plant water status). On the contrary, in the middle hours of the day, trees closed their stomata to prevent harmful water loss and this was reflected in a midday depression of transpiration fluxes, also observed in both genotypes in response to stress.

Overall, this highlights once again the limitation of stress-based measurements that do not integrate the multiple physiological responses of trees to drought stress. For example, a single measurement of stomatal conductance at 10:00 h would indicate no stress, due to the transpiration peak, while the multiple continuous measurements revealed that the high transpiration rates at that time of the day are an adaptation to reduced water availability. Hence, the use of only one indicator would have led to misleading conclusions, of difficult interpretation, as pointed out by recent studies [34].

### 4.2. Different Physiological Responses of the Two Genotypes to Water Withholding

The use of several continuous sensors was also critical to detect and understand the different physiological responses of the two genotypes to water withholding. Under stress, a more pronounced peak of transpiration was exhibited by OM. The higher morning water flux (and gas exchange) compensated for the stomatal closure later in the day, resulting in a lack of correlation between the average sap flow values and Ψ_stem_ in this cultivar (Figure 7c). Conversely, the relationship between the daily average value of sap flow density and stem water potential in NB was highly significant and positive (Figure 7c), indicating that this cultivar responded to a reduction in available water (more negative Ψ_stem_) by closing stomata and reducing water fluxes (lower *Q*_ave_). This is generally one of the first and most sensitive responses of plants to drought [35] that has been shown to vary in different olive cultivars [36].

The high rates of transpiration attained by OM are also reflected in the relatively high degree of shrinkage of both leaf and fruit tissues observed for this cultivar under drought stress. In this cultivar, a sharp drop of leaf turgor and fruit RGR was observed during the morning transpiration peak. In NB, the morning peak of sap flow was less pronounced (Figure 8a), the minimum leaf turgor measured during the day (maximum *P*_p_) was constant (Figure 4a), and the fruit RGR drop was less pronounced (Figure 5b), suggesting a very different response of this cultivar to dehydration. The analysis of the detailed daily water flux kinetics in Figure 8 highlights that the lack of turgor decrease in NB leaves during the day is due to a partial re-increase of *P*_p_ values at midday, a phenomenon called semi-inversion and associated with the decrease of turgor pressure (*P*_c_) to values close to the turgor loss point [12].

Hence, as hypothesized, the two cultivars showed different responses to water stress, including (1) a higher tendency of OM leaf tissues to shrink relative to NB, (2) a lower tendency of OM to invert the *P*_p_ curves when leaves are close to the turgor loss point than NB (Figure 3) and (3) a smaller drop in Ψ_stem_ for OM than NB (Figure 3). Such a difference in Ψ_stem_ behavior between the two cultivars has been reported in previous studies [7,13]. All these three responses of OM are typical of genotypes with more elastic tissues [37], or higher capability to adjust tissue elasticity [38], pointing to this mechanism as a potential adaptive strategy of this cultivar to cope with dehydration.

### 4.3. Cultivar Dependency of Continuous Indicator Sensitivity

The *P*_p_ readings depend on the capability of the leaf to transfer the pressure applied by the sensors’ metallic clamps, which is governed by changes in cell volume [12]. The correlation between *P*_p_ and *P*_c_ is hence based on the assumption that volume changes are directly related to *P*_c_, and tissue elastic modulus is linearly dependent on *P*_c_. This assumption implies that differences in the elastic properties of the tissues may play an important role in dictating the sensitivity of the LPCP probe to turgor, leading to the very different dynamic of maximum *P*_p_ values between NB (constant) and OM (increasing) in response to drought stress. This suggests that tissue mechanic properties should be taken into consideration in the development of irrigation strategies based on LPCP. For example, the inversion phase of the *P*_p_ curve, which allows detecting mild stress in ‘Arbequina’ [10,11], is not a good indicator of mild stress for cultivars that exhibit physiological responses to drought similar to OM.

The different drought resistance mechanisms exhibited by the two cultivars are also well depicted by the analysis of fruit RGR. While in other species, such as peach, transpiration from the cuticle has a great influence on fruit shrinkage [39], in olive, the fruits have few stomata that are rapidly covered by wax at early stages of development, hence we can assume that loss of water from the cuticle is negligible. In such conditions, we suppose that the high rate of fruit shrinkage observed in the daytime in OM is highly related to water backflow to leaves [16,40,41,42].

Interestingly, shrinking and swelling in NB fruits were very limited, and fruit RGR_range_ was a weaker predictor of mild water stress than in OM (Figure 7b). A lack of correlation between fruit RGR daily dynamics and plant water status is generally associated with a partial decrease of the hydraulic conductivity of the vascular pathway connecting the fruit and the rest of the plant, causing hydraulic isolation of the fruit [43]. However, an active accumulation of solutes, as previously observed in olive fruits [44], may also play an important role in decreasing fruit osmotic potential in NB and maintaining a more stable volume throughout the day. On the contrary, in OM the daily water flux kinetics highlights quick changes of fruit RGR in response to changes in stomatal aperture, e.g., in the morning before the transpiration peak and at midday during the decrease of transpiration fluxes, suggesting a high dependence of the fruits from the parent plant. Hence, fruit sensing seems to be a better choice in OM, since it is more representative of tree water status.

Overall, this study highlights the limitation of stress-based studies that do not consider cultivar specific drought resistance mechanisms.

## 5. Conclusions

The results of this study confirm that monitoring continuously leaf, fruit, and stem water dynamics can help detect desirable water stress levels in olive earlier than a single commonly used indicator of water stress such as the daily *P*_p_ curves. Before the *P*_p_ daily curve shape (suggested indicator of stress for ‘Arbequina’) detected any stress, trees had already developed physiological changes to deal with the reduced water availability affecting water relations between organs, and probably assimilation and growth.

In OM, fruits acted as sinks of water during the night and as water sources during the day, leading to a strong daily shrinkage and swelling pattern in response to plant stress. This highlights the important and still unrevealed role that fruits have as water storage compartments in drought resistance mechanisms of olive.

Overall, the results encourage the adoption of cultivar-specific combinations of plant-based continuously monitoring sensors for precise management of irrigation in intensive olive systems. More studies should be performed on this topic, also considering the difficulty to replicate continuous measurements on large numbers of trees. Other limitations of the sensors’ applicability to commercial orchards and their specific pro and cons have been well discussed by Fernández [45]. In particular, this study highlights that the type and magnitude of plant physiological responses to drought influence the capability of plant-based indicators to detect stress. However, such responses can be affected within the same genotype by other factors such as crop load, previously experienced stress, or phenological stage, so that further work should be done to integrate all this information in simple grower-friendly protocols, that can be developed for an efficient agricultural water management. This becomes of paramount importance in environments with reduced water availability and the need to maintain, for productive purposes, relatively high levels of plant hydration during drought-sensitive phenological stages.

## Figures and Tables

**Figure 1 plants-10-00131-f001:**
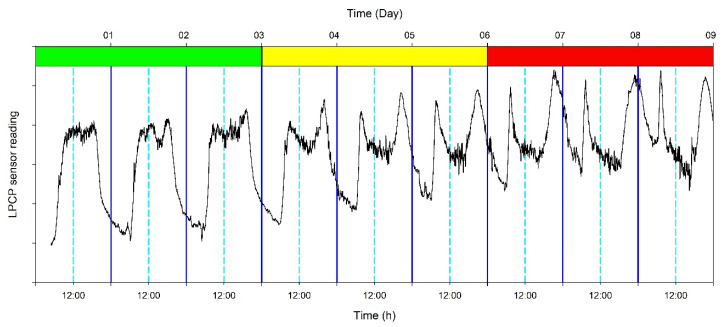
Example of the stress phases based on the inversion of the leaf patch clamp probe (LPCP) output pressure (*P*_p_) reading, with the green area showing the phase 1 or no stress (no inversion of the *P*_p_ curve, *P*_p_ equal to the inverse of turgor pressure); the yellow area indicating phase 2 or stress onset (half-inversion of the *P*_p_ curves or semi-inversion) and the red area showing phase 3 or high stress (full inversion of the *P*_p_ curve). Dark blue vertical lines indicate midnight and light blue dashed vertical lines indicate midday.

**Figure 2 plants-10-00131-f002:**
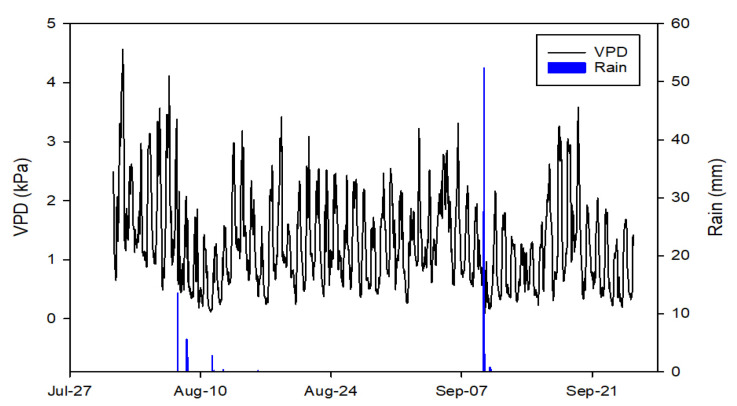
Seasonal trend of vapor pressure deficit (VPD, hourly mean) and total daily rain (from hourly values) from 1 August to 26 September 2015, obtained from data of the meteorological station in Sciacca, Sicily, Italy.

**Figure 3 plants-10-00131-f003:**
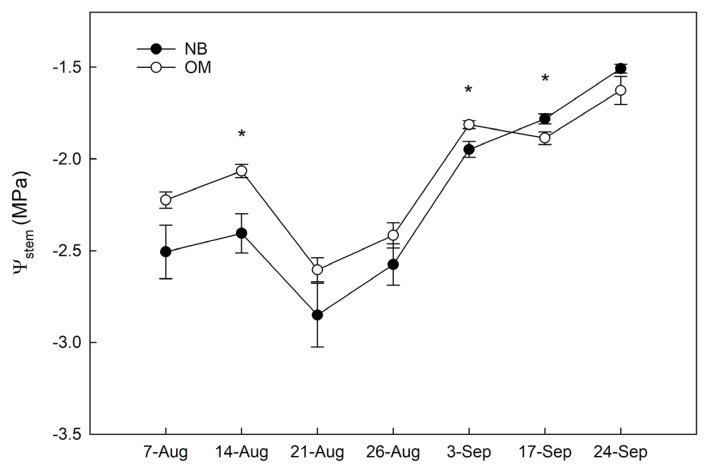
Weekly midday stem water potential (Ψ_stem_) measured for the olive cultivars Olivo di Mandanici (OM) and Nocellara del Belice (NB). Vertical black bars represent standard errors of the mean (*n* = 9); asterisks indicate significant differences between means (Tukey’s test, *p* ≤ 0.05).

**Figure 4 plants-10-00131-f004:**
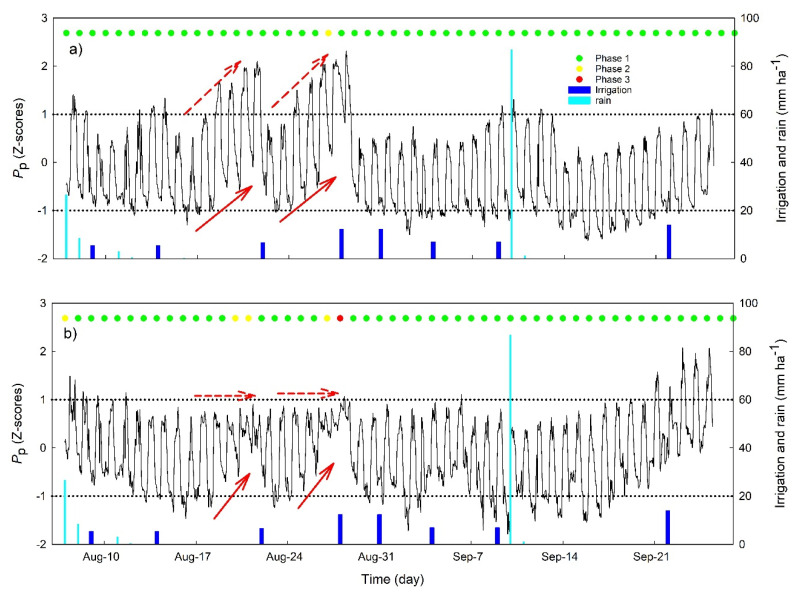
Time course of the hourly standardized leaf patch clamp pressure (LPCP) probe output pressure (*P*_p_) for the olive cultivars Olivo di Mandanici (**a**) and Nocellara del Belice (**b**). Green, yellow and red dots represent daily stress phases based on exploratory data analysis of the diurnal shape of the LPCP probe outputs: phase 1 or no stress (no inversion of the *P*_p_ curve); phase 2 or stress onset (half-inversion of the Pp curves), and phase 3 or high stress (full inversion of the *P*_p_ curve). The horizontal dotted lines are the z-score of −1 and +1, representing the thresholds within which 68.27% of the results fall. Red arrows highlight the variability of the *P*_p_ values between irrigation events discussed in the text.

**Figure 5 plants-10-00131-f005:**
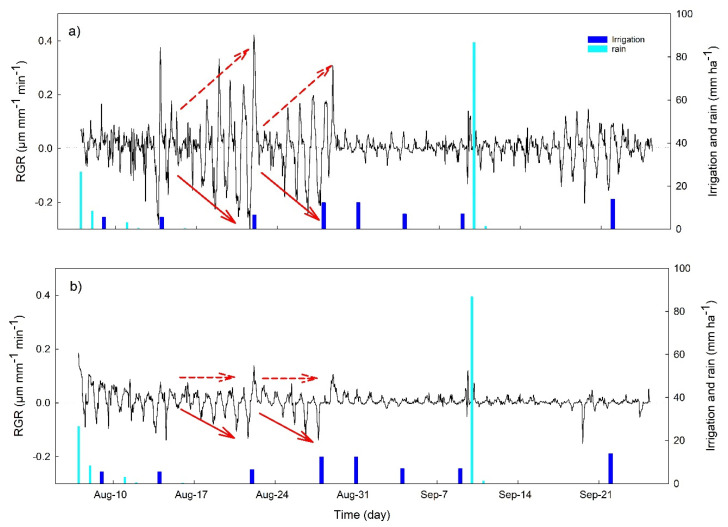
Time course of the hourly fruit relative growth rate (RGR) for the olive cultivars Olivo di Mandanici (**a**) and Nocellara del Belice (**b**). Red arrows indicate the trends of RGR peaks between consecutive irrigation events discussed in the text.

**Figure 6 plants-10-00131-f006:**
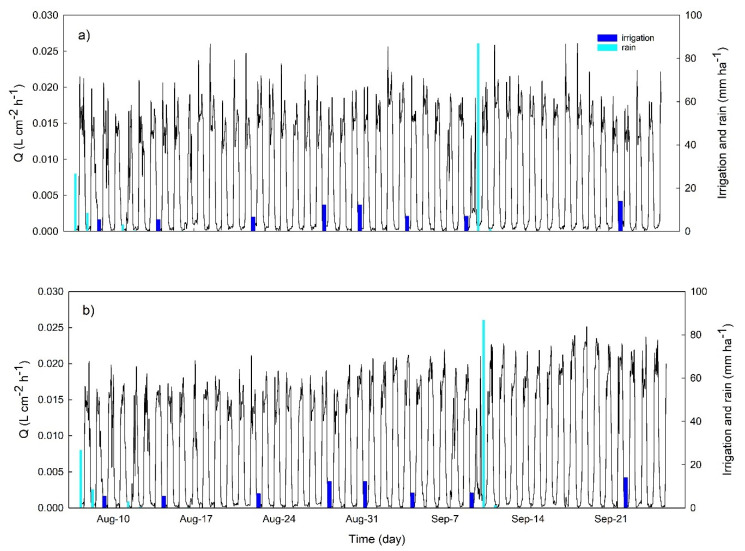
Time course of the hourly sap flow rate (*Q*) for the olive cultivars Olivo di Mandanici (**a**) and Nocellara del Belice (**b**).

**Figure 7 plants-10-00131-f007:**
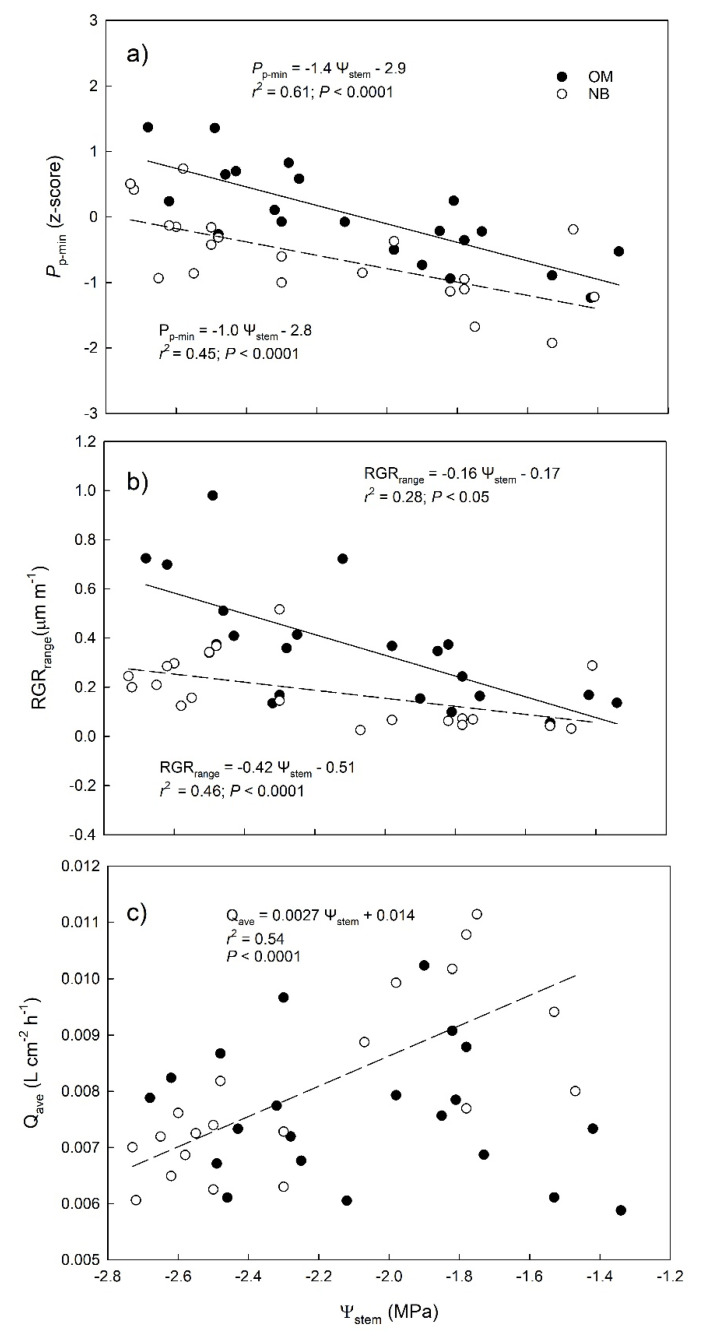
Relationships between midday stem water potential (Ψ_stem_) and the daily minimum leaf patch clamp pressure (LPCP) probe outputs value (*P*_p-min_, z-scores, panel **a**), difference between daily minimum and maximum relative growth rate (RGR_range_, panel **b**), and the daily average of hourly sap flow rate (*Q*_ave_, panel **c**) in the olive cultivars Olivo di Mandanici (OM, solid lines) and Nocellara del Belice (NB, dashed line).

**Figure 8 plants-10-00131-f008:**
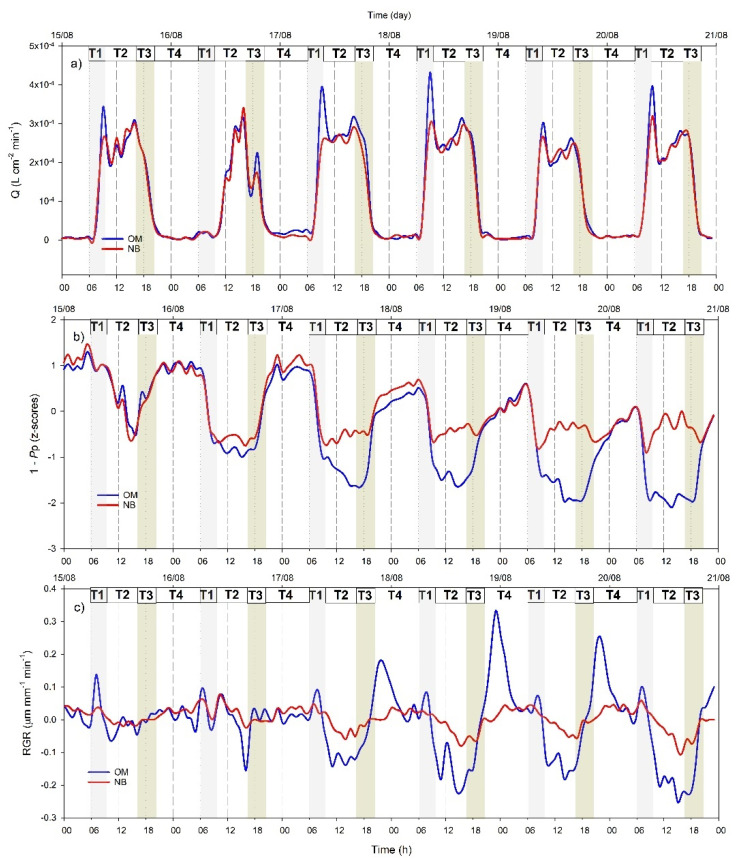
Hourly trends of (**a**) sap flow rate (*Q*), (**b**) standardized leaf patch clamp probe outputs (1 − *P*_p_) and (**c**) fruit relative growth rate (RGR) for the olive cultivars Nocellara del Belice (NB) and Olivo di Mandanici (OM) between two consecutive irrigation events, from 15 to 21 August. T1, T2 T3, and T4 represent key timeframes described in the text.

**Table 1 plants-10-00131-t001:** Comparisons of the regression slopes for the different relationships reported in Figure 6 for: daily minimum leaf patch clamp pressure (LPCP) probe outputs value (*P*_p-min_), difference between daily minimum and maximum relative growth rate (RGR_range_) or the daily average of hourly sap flow rate (*Q*_ave_) versus midday stem water potential (Ψ_stem_) between the two olive cultivars (Olivo di Mandanici and Nocellara del Belice).

Parameter	Slopes	SE	d.f.	t Ratio	*p* Value
*P* _p-min_	−0.389	0.371	38	−1.049	0.301
RGR_range_	−0.259	0.118	38	−2.195	0.034
*Q* _ave_	−0.003	0.0009	37	−3.444	0.001

## Data Availability

All data reported here is available from the authors upon request.

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
