# Peer review of "Detecting Mild Water Stress in Olive with Multiple Plant-Based Continuous Sensors"

_plants, 2021, doi:10.3390/plants10010131_

Round 1
Reviewer 1 Report
Title: Detecting mild water stress in olive with multiple plant-based continuous sensors
Confidential Comments to the EIC
The author presented a study on the sensitivity of tested indicators in two olive cultivars under water stress. Although there were some published researches on plant-based detection method of water stress, a comprehensive characterization is still needed for the sensor development. The experiment was described in great details. The statistical analysis is sufficient to allow data interpretation and results are clearly presented. In my opinion, the results are very useful, which contributes to detecting water stress level in different cultivars. There are some points needed to be minorly revised. So, I recommended publication after revision.
*Comments to the Author
Results:
L142 delete the first “probably”
L149 add “(EDA)” after “analysis”
L181 “Pp-min” changes to “Pp”
L186 “P” replaced by “P”
L210 Please add explain of “Pc”
Discussion:
L253-255 According to Figure 6c, the relationship between the daily average value of sap flow density and stem water potential in NB was positive, not “negative”. Please add explain and reference.
L258 change “Figure 4” to “Figure 5”
L275-278 Please rephrase this sentence.
Please enhance the water stress response part by citing some current papers.
S. Li, X. Li, Z. Wei, F. Liu, ABA-mediated modulation of elevated CO2 on stomatal response to drought, Current Opinion in Plant Biology, 56 (2020) 174-180.
M&M:
L386 delete “,”
Author Response
*Comments to the Author
Results:
L142 delete the first “probably”
Done
L149 add “(EDA)” after “analysis”
Added
L181 “Pp-min” changes to “Pp”
Done
L186 “P” replaced by “P”
Done
L210 Please add explain of “Pc”
Done
Discussion:
L253-255 According to Figure 6c, the relationship between the daily average value of sap flow density and stem water potential in NB was positive, not “negative”. Please add explain and reference.
Thank you for getting the mistake obviously caused by the negative nature of the SWP. We changed negative with positive and added a more detailed description of the relative change in the parameters we refer to, to clarify and avoid doubts (lines 328-331)
L258 change “Figure 4” to “Figure 5”
Done
L275-278 Please rephrase this sentence.
We rephrased and simplified the sentence. We hope it is clearer now. (lines 344-349)
Please enhance the water stress response part by citing some current papers.
S. Li, X. Li, Z. Wei, F. Liu, ABA-mediated modulation of elevated CO2 on stomatal response to drought, Current Opinion in Plant Biology, 56 (2020) 174-180.
We added the suggested and some extra citations to better explain the response to drought:
Zhang, X., Wang, Y., Huang, G., Feng, F., Liu, X., Guo, R., ... & Mei, X. (2020). Atmospheric humidity and genotype are key determinants of the diurnal stomatal conductance pattern. Journal of Agronomy and Crop Science, 206(2), 161-168.
Gucci, R., Grimelli, A., Costagli, G., Tognetti, R., Minnocci, A., & Vitagliano, C. (2000, September). Stomatal characteristics of two olive cultivars" Frantoio" and" Leccino". In IV International Symposium on Olive Growing 586 (pp. 541-544).
S. Li, X. Li, Z. Wei, F. Liu, ABA-mediated modulation of elevated CO2 on stomatal response to drought, Current Opinion in Plant Biology, 56 (2020) 174-180.
M&M:
L386 delete “,”
Done
Attached is a pdf copy of the manuscript with track changes highlighting the modifications applied in response to all the Reviewers

Reviewer 2 Report
This is a well written and well presented study showing important data regarding detailed cultivar responses and the importance of continuously monitoring physiological parameters. It is disappointing only three trees per cultivar and two cultivars could be included but more data is always going to be more desirable.
With minor corrections, in my opinion the paper will be ready for publication.
The experimental design should be clarified, how were the six trees arranged and blocked?
Please also clarify sensor numbers, three trees per cultivar each sensor on each tree?
Full names for the Cultivars OM & NB need defining at the first use of the abbreviation on page 4. (This definition should not occur first in a figure caption.)
Current knowledge of the cultivars drought tolerance would be a valuable inclusion to the introduction. As would visual example of Pp daily curve, no drought stress, moderate and inverted were discussed, could the author include this as a graphical example?
I have many concerns with the methods section being included after the discussion but I assume this is the journals format. Introducing the cultivars studied in the intro will help significantly.
Small edits for in text:
p3 line 89: economic (not economical)
p11 line 278: what has been reported
p13 line 369: Granier-type heat dissipation probes
Author Response
With minor corrections, in my opinion the paper will be ready for publication.
The experimental design should be clarified, how were the six trees arranged and blocked?
We improved description of the experimental design (lines 430-435), it should be clear now.
Please also clarify sensor numbers, three trees per cultivar each sensor on each tree?
Better explained in lines 434-435
Full names for the Cultivars OM & NB need defining at the first use of the abbreviation on page 4. (This definition should not occur first in a figure caption.)
Done
Current knowledge of the cultivars drought tolerance would be a valuable inclusion to the introduction. As would visual example of Pp daily curve, no drought stress, moderate and inverted were discussed, could the author include this as a graphical example?
Done in lines 108-110 and figure 1
I have many concerns with the methods section being included after the discussion but I assume this is the journals format. Introducing the cultivars studied in the intro will help significantly.
Yes, this is the journal format; we improved the introduction to drive the reader from intro to results with as much information as possible
Small edits for in text:
p3 line 89: economic (not economical)
p11 line 278: what has been reported
p13 line 369: Granier-type heat dissipation probes
All addressed
Attached is a pdf copy of the manuscript including all changes in response to all the Reviewers

Reviewer 3 Report
An interesting study which deals with the reliable detection of the mild stress based on the monitoring of the several physiological parameters in olive trees. However, the results are presented very descriptively. The experimental design should be given in more detail. The conclusions lack a clear definition of the obtained findings and definition of the possible issues of the presented study. For example, what are the limits in application of tested parameters for the detection of water stress? What other issues should be addressed by research in this area. The manuscript requires several changes and more concise presentation of the results and findings.
- The chapter Material and Methods is usually before the Results. The manuscript should be rearranged.
- The experimental design should be explained in more detail. Define was the position of the sampled branches within a tree etc. Is the sample of 3 trees reliable for screening of the physiological parameters? Was the minimum sample size test performed?
- Chapter 2.1 Environmental conditions should be moved under Material and Methods.
- Chapter 3.2 should have a more concise title, as the aim of the presented research is different from monitoring.
- The results must include basic knowledge, which will be discussed further. Now, the results describe mainly the changes in the physiological parameters. but in the current form do not lead to the confirmation or rejection of the hypotheses. The presentation of the results should be less descriptive and more factual.
- In Figure 6, the regression equations, r2 and P values should be given in the graphs.
- In Discussion, the findings and facts obtained within the study should be discussed. Therefore, remove the links (citations) for tables and figures already mentioned in Results, be consistent. The discussion should be improved.
- The conclusions lack a clear definition of the obtained findings and definition of the possible issues of the presented study.
Author Response
An interesting study which deals with the reliable detection of the mild stress based on the monitoring of the several physiological parameters in olive trees. However, the results are presented very descriptively. The experimental design should be given in more detail. The conclusions lack a clear definition of the obtained findings and definition of the possible issues of the presented study. For example, what are the limits in application of tested parameters for the detection of water stress?
Other papers have been published with the aim of identifying pros and cons of these plant based sensors for water management (Fernandez 2014, https://doi.org/10.1016/j.agwat.2014.04.017), so we believe further discussing this would be out of the scope (and objective) of this specific paper, but we cited Fernandez’s paper for reference in the conclusions.
What other issues should be addressed by research in this area.
Discussion of potential future research was added in the conclusion.
The manuscript requires several changes and more concise presentation of the results and findings.
-
The chapter Material and Methods is usually before the Results. The manuscript should be rearranged.
Placing Material and Methods after the Results is a requirement from the Journal.
-
The experimental design should be explained in more detail. Define was the position of the sampled branches within a tree etc. Is the sample of 3 trees reliable for screening of the physiological parameters? Was the minimum sample size test performed?
We improved description of placement of sensors within the trees (lines 449-450, 472-473), and clarified the experimental design (lines 430-435). We are aware that higher replication of the continuous measurements would be desirable, but having three replication trees is quite common in sensor-based experiment, due to the still high costs associated to these technologies affecting the whole economy of the investigation and ultimately limiting replications.
We have carried out other studies with the same continuous plant based sensors and on similar settings; similar experimental layouts have proven to be suitable to reveal the investigated physiological mechanisms. Following some examples:
Scalisi, A.; Marino, G.; Marra, F.P.; Caruso, T.; Lo Bianco, R. A Cultivar-Sensitive Approach for the Continuous Monitoring of Olive (Olea europaea L.) Tree Water Status by Fruit and Leaf Sensing. Front. Plant Sci. 2020, doi:10.3389/fpls.2020.00340.
Marino, G.; Pernice, F.; Marra, F.P.; Caruso, T. Validation of an online system for the continuous monitoring of tree water status for sustainable irrigation managements in olive (Olea europaea L.). Agric. Water Manag. 2016, doi:10.1016/j.agwat.2016.08.010.
Scalisi, A.; Marra, F.P.; Caruso, T.; Illuminati, C.; Costa, F.; Lo Bianco, R. Transpiration rates and hydraulic conductance of two olive genotypes with different sensitivity to drought. In Proceedings of the Acta Horticulturae; 2019.
Scalisi, A.; O’Connell, M.G.; Stefanelli, D.; Lo Bianco, R. Fruit and leaf sensing for continuous detection of nectarine water status. Front. Plant Sci. 2019, doi:10.3389/fpls.2019.00805.
Grilo, F.S., Scalisi, A., Pernice, F., Morandi, B., Lo Bianco, R. (2019). Recurrent deficit irrigation and fruit harvest affect tree water relations and fruitlet growth in 'Valencia' orange. Europ. J. Hort. Sci. 84(3):177–187 doi: 10.17660/eJHS.2019/84.3.8.
Also, many of the previous works from different Authors on continuous plant-based monitoring use a similar numbers of replicate trees, i.e. three trees or less. Following some examples:
Padilla Diaz et al 2015 (http://dx.doi.org/10.1016/j.agwat.2015.08.002), in one of the main paper about the use of the LPCP in olive, report: “Both in the FI and 45RDI treatments, and before the beginning of the 2013 and 2014 irrigation seasons, one central tree per plot was instrumented, in three of the four plots, with ZIM probes (YARAZIM Plant Technology, Hennigsdorf, Germany).”
Rodriguez et al 2012 (http://dx.doi.org/10.1016/j.agwat.2012.07.007) comparing Sap flow and LPCP in olive report : ‘we instrumented one tree per plot, in three plots per RDI treatment. In the control plot we instrumented three trees with 2 zim probes per tree.
Garcia tejero et al 2012 (https://doi.org/10.1016/j.scienta.2011.10.022) develop a relation between fruit shrinkage and trunk shrinkage measured fruit growth rate on a larger sample (“Fruit-growth rates were determined by carefully selecting 8 fruits from nearly equal-sized spurs well distributed around the tree canopy from four replicates (32 fruits per treatment). However the trunk diameter replication was still low: “On the other hand, micrometric trunk-diameter fluctuations (TDF) were measured on three trees per treatment, in 15-min intervals, using a set of PlantSens sensor (CPS Factory, Verdtech Nuevo Campo, SA, Spain), which registers the variations via measuring sensor”
Ehrenberger et al 2012, (doi:10.1111/j.1438-8677.2011.00545.x) in the main paper on the use of zim probes, where the inversion staged for stress detection were firstly described report results measured on one single tree.
We performed the stem water potential measurements on a larger sample size (all the trees in the block, for a total of 9 trees per date and cultivar), but since data on water status were very similar among the selected trees, we thought that for the scope of the paper was sufficient to show only the SWP of the three monitored trees. However, considering the funded doubt of the reviewer, we substituted Fig. 2 with a new SWP graph showing means and standard errors of the bigger population (9 trees). We also added this aspect as a potential limitation of this type of studies in the Conclusions.
-
Chapter 2.1 Environmental conditions should be moved under Material and Methods.
Moved
-
Chapter 3.2 should have a more concise title, as the aim of the presented research is different from monitoring.
changed
-
The results must include basic knowledge, which will be discussed further. Now, the results describe mainly the changes in the physiological parameters. but in the current form do not lead to the confirmation or rejection of the hypotheses. The presentation of the results should be less descriptive and more factual.
We made some changes in the Result section that improved its readability, as suggested. Specifically, we added new subheadings and introductory paragraphs associated with our hypotheses. In addition to providing information about the confirmation of the hypotheses, these changes allowed a clear separation of the more factual and descriptive parts. In this regard, we are aware that the last part of the Results, namely the last subheading, is mainly descriptive. However, we consider it is important to provide this detailed analysis to better understand both the results presented before and the Discussion (which we aim to be concise).
-
In Figure 6, the regression equations, r2 and P values should be given in the graphs.
Done
-
In Discussion, the findings and facts obtained within the study should be discussed. Therefore, remove the links (citations) for tables and figures already mentioned in Results, be consistent. The discussion should be improved.
We substantially modified the Discussion for a better clarity, rearranging the text and including new subheadings that are related to our hypotheses. We believe these changes significantly improved the Discussion. We kept the references to the Tables and Figures when we considered it was helpful to guide the reader through the text. Citing Tables or Figures is indeed a common practice in the majority of articles with separate Results and Discussion sections to help the reader follow the discussion.
-
The conclusions lack a clear definition of the obtained findings and definition of the possible issues of the presented study.
In response to the comment, we highlighted the conclusions of the paper throughout the Discussion section and also modified the Conclusion section.
As a result, the Conclusions are now complete since they include:
-
Clear statement of the confirmation of the hypothesis
-
Pitfalls of the study
-
Potential future research
-
Implication for stakeholders
Attached is a pdf copy of the manuscript including all changes in response to all the Reviewers

Round 2
Reviewer 1 Report
All comments have been considered.
Reviewer 2 Report
Thank you for making the suggested edits these have improved the paper and answered the questions I had.
If you could clarify (or remove) the sentence "Measurements were performed in the central tree of each block" (line 420)
The next line:
"One sensor of each type was installed in each selected tree, for a total 18 sensors (three sensors types per three trees per two cultivars)."
My understanding of the experimental design is you have three blocks for each cultivar, each cultivar in a different row? 9 trees were measured per cultivar. = 18 sensors. So line 420 is confusing - were you only measuring the central tree in each block? or was this only Ψstem was measured on the central tree?
Reviewer 3 Report
Reviewer's comments 2nd round
The manuscript was improved, but there are still suggested following changes:
- The paper is descriptive, so it is necessary to present the Material and Methods before the Results. It is important to know HOW the research was carried out before the research outputs and findings are presented to the readers.
The journal Plants accepts free format submission, as it is declared under instructions for authors: „We do not have strict formatting requirements, but all manuscripts must contain the required sections: Author Information, Abstract, Keywords, Introduction, Materials & Methods, Results, Conclusions, Figures and Tables with Captions, Funding Information, Author Contributions, Conflict of Interest and other Ethics Statements. Check the Journal Instructions for Authors for more details“ https://www.mdpi.com/journal/plants/instructions
- It must be clearly stated in the experimental design (with reference to the previous work of the authors) that the scope of the sample and the number of measurements correspond to the standards for this type of research.The conditions in which the research is carried out, as well as the scope of the obtained data support the strength of the presented outputs.These infomration must be supplemented in the manuscript.

Author Response
The manuscript was improved, but there are still suggested following changes:
1) The paper is descriptive, so it is necessary to present the Material and Methods before the Results. It is important to know HOW the research was carried out before the research outputs and findings are presented to the readers.
The journal Plants accepts free format submission, as it is declared under instructions for authors: „We do not have strict formatting requirements, but all manuscripts must contain the required sections: Author Information, Abstract, Keywords, Introduction, Materials & Methods, Results, Conclusions, Figures and Tables with Captions, Funding Information, Author Contributions, Conflict of Interest and other Ethics Statements. Check the Journal Instructions for Authors for more details“ https://www.mdpi.com/journal/plants/instructions
We decided to follow your recommendation, which makes good sense and as we gladly learned there should be no conflict with the Journal format.
2) It must be clearly stated in the experimental design (with reference to the previous work of the authors) that the scope of the sample and the number of measurements correspond to the standards for this type of research. The conditions in which the research is carried out, as well as the scope of the obtained data support the strength of the presented outputs. These infomration must be supplemented in the manuscript.
We added a sentence (lines 156-158) stating clearly what you suggest and citing previous works with similar experimental layout.
Round 3
Reviewer 3 Report
Accepted without other comments.